# Supervised score aggregation for active anomaly detection

**Kevin Bleakley**                                                    *kevin.bleakley@inria.fr*
*Inria and Laboratoire de Mathématiques d'Orsay, France*

**Mouhcine Mendil**                                        *mouhcine.mendil@irt-saintexupery.com*
*IRT Saint Exupéry, France*

**Martin Royer**                                                    *martin.royer@irt-systemx.fr*
*IRT SystemX, France*

**Benjamin Auder**                                        *benjamin.auder@universite-paris-saclay.fr*
*Laboratoire de Mathématiques d'Orsay, France*

**Reviewed on OpenReview:** *https: // openreview. net/ forum? id= nrmJD3XMA3*

## Abstract

Detecting rare anomalies in batches of multidimensional data is challenging. We propose an original supervised active-learning framework that sends a small number of data points from each batch to an expert for labeling as 'anomaly' or 'nominal' via two mechanisms: (i) points most likely to be anomalies in the eyes of a supervised classifier trained on previously-labeled data; and (ii) points suggested by an active learner. Instead of training the supervised classifier directly on currently-labeled raw data, we treat the scores calculated by an ensemble of $M$ user-defined unsupervised anomaly detectors as if they were the learner's input features. Our approach generalizes earlier attempts to linearly aggregate unsupervised anomaly detector scores, and broadens the scope of these methods from unordered bags of data to ordered data such as time series. Simulated and real data trials show that this method outperforms—often significantly—linear strategies. The Python library `acanag` implements our proposed method.

## 1 Introduction

### 1.1 General setting

Anomaly detection in large multidimensional data sets is difficult when anomalies are rare, and few (or zero) true anomalies have been previously labeled. In order to deal with situations like this, anomaly detection methods have evolved over time from an original focus on unsupervised strategies to more sophisticated techniques today.

Many previous studies have considered anomaly detection in unordered bags of data (Islam et al., 2018; Pevnỳ, 2016). A common statistical assumption in such settings is that individual data are i.i.d. (independent and identically distributed) samples from a mixture distribution of "normal" (*nominal*) and "abnormal" (*anomaly*) data with density $f = \tau f_{anom} + (1-\tau)f_{nom}$. Here, $\tau$ is the probability of drawing an anomaly (from $f_{anom}$) and $(1-\tau)$ the probability of drawing a nominal (from $f_{nom}$) (Pimentel et al., 2020). However, this i.i.d. mixture model does not capture more general scenarios. For instance, an anomaly may not solely be a function of its raw data value; its status may instead depend on its value *in the context of* other raw data values surrounding it in an unordered bag or ordered time series. For example, in a non-stationary time

series, anomalies might be related to abrupt changes between successive time points rather than directly to raw data values.

In this paper we propose a general method to detect anomalies in batches of ordered or unordered data using unsupervised anomaly detectors, supervised learning and active learning. Our method builds on previous developments in unsupervised anomaly detection (Pevnỳ, 2016; Das et al., 2016; Islam et al., 2018; Williams et al., 2002; Schölkopf et al., 1999; He et al., 2003; Liu et al., 2012), aggregation of anomaly detectors (Benferhat & Tabia, 2008; Gao & Tan, 2006; Gao et al., 2012; Kriegel et al., 2011; Kruegel & Vigna, 2003; Kruegel et al., 2003), supervised learning (Almgren & Jonsson, 2004; Chapelle et al., 2006), and active learning (Balcan et al., 2007; Dasgupta et al., 2005; Freund et al., 1997; Roy & McCallum, 2001; Das et al., 2016; Danka & Horvath, 2018; Bodor et al., 2022; Das et al., 2017; Islam et al., 2018; Pimentel et al., 2020; Siddiqui et al., 2018; Tang et al., 2020; Görnitz et al., 2013). Please refer to the Appendix for a more detailed introduction to these methods.

## 1.2 Why the problem matters

Rare anomaly detection in high-dimensional datasets is a *very hard problem* yet also important in a modern world increasingly overwhelmed by data. Critical applications include detecting network intrusion (Yamanishi et al., 2000), analyzing satellite images for unexpected features (Asuncion et al., 2007), and looking for anomalies in mammography images (Rocke & Woodruff, 1996).

## 1.3 Main contribution and intuition

Individual unsupervised anomaly detectors are typically designed in the "hope" that true anomalies get the highest scores and are thus detectable. Some previous methods aggregate such scores from ensembles of unsupervised anomaly detectors using linear combinations (Pevnỳ, 2016; Das et al., 2016; Islam et al., 2018), since aggregation can potentially improve overall anomaly detection. Aggregation can be done in an unsupervised way (e.g., mean score) or a supervised way (e.g., learn best linear weights) if some labeled data is available. However, if true anomalies actually have "med-range" scores while nominals have lower and higher scores, linear methods simply cannot map anomalies to the highest values.

Our main contribution is to directly treat ensemble anomaly scores for data batches as if they were data from a distribution. We then use labeled scores (obtained via the data) to train a supervised classifier to predict new anomalies in new data batches. This is a direct generalization of linear methods. Since in real-world settings it can be expensive to check large numbers of predictions, we also use active learning to provide a small number of candidates from each new batch to an expert for labeling. We prove that under certain hypotheses, pooled anomaly scores and pooled nominal scores have probability distributions. Hence, supervised learning on scores and labels is well-defined.

## 1.4 Paper structure

In Section 2 we present our method in detail and describe its similarities and differences to previous methods. In Section 3 we test it on simulated and real data sets and compare it to three other methods. In Section 4 we conclude and provide perspectives for future work.

## 2 Methods

### 2.1 Description of the method

A simplified flowchart describing the AAA or *Active Anomaly Aggregation* method is presented in Fig. 1, while Algorithm 1 gives the full pseudo-code.

The key idea is that unsupervised anomaly detectors whose *score distributions* (see Section 2.4) for true anomalies turns out to be "different enough" from those of nominals can potentially provide discriminative information to a supervised classifier. The AAA method requires three main *a priori* choices: (1) an ensemble of $M$ unsupervised anomaly detectors, (2) a supervised classifier $\mathcal{C}$, and (3) an active learning strategy $\mathcal{A}$.

Fig. 1 presents the basic process: receive a new data batch, calculate its anomaly scores, predict the most likely anomaly candidates in that batch using the current state of the supervised classifier (trained on all scores associated with previously-labeled data), add a small number of extra candidates via active learning, give all candidates to an expert for labeling, and then retrain the classifier on all currently-labeled data's scores.

## 2.2 Original features of the approach

To the best of our knowledge, our method is the first to combine anomaly scores from ensembles of unsupervised anomaly detectors, supervised learning *on these anomaly scores*, and active learning. In particular, using score vectors as if they were training features seems conceptually new in the general setting, though hints of this appear in specific settings for other methods (see Section 2.3).

An implicit hypothesis underlies most anomaly detection methods in the unsupervised scores setting: you need true anomalies to get the highest scores in order to be detected. It is thus important to emphasize that, unlike previous methods (e.g., Das et al. (2016) or Pevnỳ (2016)), the proposed method does *not* require anomalies to obtain the highest (or lowest) scores from at least one anomaly detector in an ensemble. Instead, any consistent difference in score distributions between anomalies and nominals can be leveraged by the classifier, regardless of whether the scores are high, low, or in-between.

## 2.3 Comparisons with previous work

Comparisons here are for the most part chronological. In Kruegel et al. (2003); Kruegel & Vigna (2003); Gao & Tan (2006); Benferhat & Tabia (2008); Kriegel et al. (2011), various unsupervised schemes were presented to aggregate anomaly scores from an ensemble using Bayesian techniques to hopefully push linear combinations of scores for true anomalies higher than those for nominals, but with all of the usual defaults of purely unsupervised anomaly detection.

The ALADIN algorithm (Stokes et al., 2008) involved active learning on the original data itself rather than on anomaly scores, training a supervised classifier to learn a decision boundary that separated anomalies and nominals based on labeled data. The SSAD algorithm in Görnitz et al. (2013) performed active learning with supervised or semi-supervised learning for anomaly detection on the original data using support vector data descriptions (Tax & Duin, 2004) and kernel functions, but without an ensemble of anomaly detectors. In Rabenoro et al. (2014)—in an original twist—the authors generated a large number of anomaly detectors, not only by having an ensemble of different models, but also by including the same model, with *different* parameter values, as separate models. They did not use active learning to improve aggregation over time.

A random projection strategy called LODA was used in Pevnỳ (2016) to create an ensemble of detectors. In it, each anomaly detector corresponded to a randomly generated sparse $d$-dimensional vector of real numbers, mapping the $d$-dimensional data to $\mathbb{R}$ by taking the scalar product. An empirical distribution in $\mathbb{R}$, represented as an equal-bin-width histogram, could then be formed for the random projector by applying it to the full data set. The motivation was that projected data points in low density areas of such histograms should be more likely, on average across the set of random projections, to correspond to true anomalies. They therefore defined the score as the negative log of each projected point's empirical bin density in order to hopefully push anomalies' scores to the top. Some geometric intuition on LODA—provided in the Appendix—partly explains why LODA can fail completely in some settings (see Section 3).

Since LODA is unsupervised and constrained to use the arithmetic mean score across projectors as the final score, the idea was extended to active-LODA (Das et al., 2016): greedy active learning with a linear classifier, optimizing ensemble weights in such a way that labeled true anomalies are pushed toward higher real numbers than labeled true nominals. In contrast, our AAA method is not constrained to inflexible hyperplanes as decision boundaries induced by linear combinations when aggregating scores. Note that due to its pairwise optimization constraints between all currently labeled-data, active-LODA struggles to converge once there are hundreds of labeled data points.

Some issues with active-LODA were taken into account in the GLAD algorithm (Islam et al., 2018) using LODA projections as anomaly detectors in parallel with a neural network. Instead of the weights being

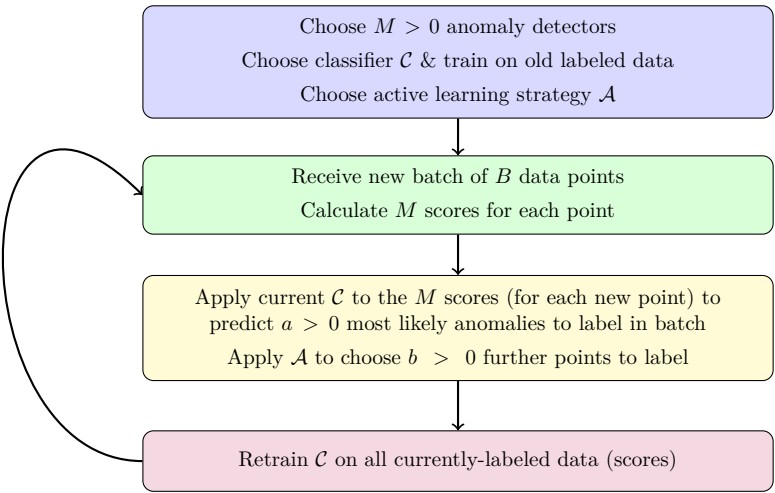

Figure 1: Simplified schematic showing how Active Anomaly Aggregation works.

---

**Algorithm 1** AAA: Supervised score aggregation for active anomaly detection

---

**Initialization:**
Start with either (i) no old data or (ii) $T$ (possibly time-ordered) $d$-dimensional real-valued data points $X_1, \ldots, X_T$ stored as a $T \times d$ matrix $\mathbf{X}_{old}$. Assume either (a) none of the data is labeled, (b) some data is labeled, or (c) all data is labeled. Choose a supervised classifier $\mathcal{C}$. Choose an active learning strategy $\mathcal{A}$. Choose $M$ unsupervised anomaly detectors $\{U_1, \ldots, U_M\}$. Choose an integer $r > 0$. Choose an integer $D > 0$.

**if** (i) or (ii & a) **then**
    initialize $\mathbf{S}_{curr}$ as an empty matrix and $Y_{curr}$ as an empty vector.
**else**
    Calculate anomaly scores for $\mathbf{X}_{old}$ for each of the $M$ anomaly detectors at each of the $T$ data points, giving a $T \times M$ score matrix $\mathbf{S}_{old}$. Extract rows of $\mathbf{S}_{old}$ corresponding to rows of $\mathbf{X}_{old}$ which had labels. Call this submatrix $\mathbf{S}_{curr}$ and put the corresponding labels in a vector $Y_{curr}$.
    **if** there are at least $r$ labeled data points from each class **then**
        train $\mathcal{C}$ on $\mathbf{S}_{curr}$ and $Y_{curr}$.
    **end if**
**end if**

**Iteration:**
For iteration $i \geq 1$, receive a batch of $B_i \leq D$ new data points $\mathbf{X}_{new}$. Concatenate $\mathbf{X}_{new}$ onto the bottom of $\mathbf{X}_{curr}$. Extract the last $D$ rows and replace $\mathbf{X}_{curr}$ by them.

**if** there are not at least $r$ labeled data points from each class **then**
    Randomly select $q_i$ queries ($0 < q_i \leq B_i$) from the new batch only. Send them to the expert for labeling, thus obtaining $q_i$ labels as a vector $Y_{new}$. Update $Y_{curr}$ by concatenating $Y_{new}$ on to the end of it. Calculate the $D \times M$ matrix of scores for $\mathbf{X}_{curr}$ and extract from it the rows corresponding to the $q_i$ queries; call this submatrix $S_{new}$. Update $S_{curr}$ by concatenating $S_{new}$ on to the end of it.
    **if** there are now at least $r$ labeled data points from each class **then**
        train $\mathcal{C}$ on $\mathbf{S}_{curr}$ and $Y_{curr}$.
    **end if**
**else**
    Calculate the score matrix for $\mathbf{X}_{curr}$ and retain the last $B_i$ rows; call this $\mathbf{S}_{next}$. Apply $\mathcal{C}$ to it to predict labels. Choose $r_i$ queries ($0 \leq r_i \leq B_i$) based on the predicted labels and send them to the expert. Apply $\mathcal{A}$ to $\mathbf{S}_{next}$ with respect to $\mathcal{C}$ and, based on this, choose $q_i$ queries ($0 < q_i \leq B_i$) for the expert. Remove any duplicate queries and send the remaining ones to the expert. Update $S_{curr}$ by concatenating all queried rows of $\mathbf{S}_{next}$ on to it. Update $Y_{curr}$ by concatenating all newly obtained labels on to it. Retrain $\mathcal{C}$ on $S_{curr}$ and $Y_{curr}$.
**end if**

---

determined via a loss function for weighted linear combinations—and requiring identical weights to be applied to all input data, they learned "local" weights by—in parallel to performing projections—passing each original data point through a neural network trained to output "good" weights, on average, for each currently-labeled data point. Since these output weights were now not the same for different data points, this algorithm corresponded to a kind of 'local correction' to underlying weighted linear combination learning. It nevertheless required learning thousands of parameters of a neural network simply to output linear combination weights, often with tiny amounts of labeled data to begin with—not necessarily ideal when anomalies are rare. Furthermore, if anomalies are highly contextual (e.g., a function of their relationship to other data points rather than of their individual values), GLAD can fail completely (see Section 3). In contrast to this unwieldy idea, the AAA method is simple: Apply a user-selected trained supervised learner *directly to the anomaly score vector* output by an ensemble of anomaly detectors in order to predict anomalies, with the additional assistance of an active learner.

Since these three originally LODA projection-based strategies described above Das et al. (2016); Islam et al. (2018); Pevnỳ (2016) span the range from unsupervised aggregation up to the use of neural networks, and furthermore can be directly applied to any ensemble of anomaly detectors (i.e., not just LODA projections), we chose to compare their performance with AAA on a range of simulated and real data settings in Section 3.

For completeness, we briefly mention a few other related methods. In (Das et al., 2017; 2018; Siddiqui et al., 2018), the detector ensemble was made up specifically of different isolation forest trees but still involved learning linear combinations of scores, while in Tang et al. (2020) the authors used arbitrary anomaly detectors in the ensemble, but still in parallel trained a neural network on the original data to learn weights in order to perform "local" linear combinations of the scores like in Islam et al. (2018). In Veeramachaneni et al. (2016) the authors used an ensemble of anomaly detectors to output scores, then used non-adaptive rules to provide queries to an expert; they then trained a classifier on the subset of the original data which was now labeled (i.e., not on the scores). Lastly, in Xin et al. (2023) the authors specifically trained a neural network directly on the scores output by an ensemble. However, they pre-processed the data using PCA and did not use active learning, instead supposing a fully-labeled data set—implausible in real-world settings.

## 2.4 The elephant in the room: Score distributions

So far we have implicitly assumed that the "score distributions" we treat as input features are either mathematically well-defined, or at least useful in practice. We now take a closer look at this issue. Here, each of the $M$ anomaly score functions maps a bag or batch of $n$ $d$-dimensional data points $\{X_1, \ldots, X_n\}$ to $n$ scores in $\mathbb{R}$. A successful supervised classifier acting on these scores would require (1) some kind of stability in the scores and (2) "separation" between typical scores assigned to true anomalies and true nominals, in order to learn well. A potential spanner in the works is that, for example, even in the simple case of one fixed LODA projection as the score function (see Section 2.3), the obtained score for any data point *depends on the values of the other $n-1$ data points in the bag/batch*, due to LODA's deterministic histogram binning process; see Pevnỳ (2016) and Birgé & Rozenholc (2006). The same issue affects other contextual anomaly detectors like isolation forest and local outlier factor (Liu et al., 2012; He et al., 2003).

More precisely, it is not intuitively clear that by repeatedly drawing new bags or batches of size $n$, pooled scores for true nominals and true anomalies will act as if generated by probability distributions ("score distributions"). The natural question is therefore: *Do there exist conditions under which "score distributions" correspond to actual probability distributions?* We do not have a general answer to this question, but can prove some preliminary results in the LODA projection setting.

Consider the LODA projection setting (Section 2.3) and i.i.d. $d$-dimensional data arriving in batches $X^{(j)}$, $j = 1, 2, \ldots$, of fixed size $n \geq 1$:

$$X^{(j)} = (X_1^{(j)}, \ldots, X_n^{(j)}) \in (\mathbb{R}^d)^n$$

generated from a mixture distribution

$$\mu = (1 - \tau)\mu_{\text{nom}} + \tau\mu_{\text{anom}}, \qquad \tau \in ]0, 1[,$$

where $\mu_{\text{nom}}$ and $\mu_{\text{anom}}$ are probability laws on $\mathbb{R}^d$ for nominal and anomalous data, respectively, and between-batch data is also i.i.d. Following the LODA method in the one-dimensional setting, we draw a random

projection vector $L \in \mathbb{R}^d$ (independent of all $X_i^{(j)}$) by generating i.i.d. $N(0,1)$ entries and randomly setting all but $\lceil \sqrt{d} \rceil$ coordinates to zero, where $\lceil \cdot \rceil$ denotes the ceiling function; $L$ is then fixed at that value. Next, we define a measurable transform $\mathcal{T}$, corresponding to calculating LODA scores on the $j$th batch, as follows:

1. Map each $X_i^{(j)}$ to a real number $Y_i^{(j)} = L^T X_i^{(j)}$.

2. Within the batch, run a deterministic algorithm $\mathcal{D}$ (see (Birgé & Rozenholc, 2006)) that chooses an optimal number of equal-length bins $b_{\text{best}} \in \{1, \ldots, n\}$ based on the values $\{Y_i^{(j)}\}$, then assigns to each $Y_i^{(j)}$ a value

$$R_i^{(j)} = -\log \left( \text{empirical frequency of the bin containing } Y_i^{(j)} \right).$$

Then, let $K_j$ denote the number of true anomalies in the batch; in the current setting, $K_j \sim \text{Bin}(n, \tau)$. If $K_j = 0$, then the batch contributes nothing to the pooled anomaly scores from the previous $j - 1$ batches. Otherwise, let

$$R_{\text{anom},1}^{(j)}, \ldots, R_{\text{anom},K_j}^{(j)}$$

be the corresponding anomaly scores. Define the empirical cumulative distribution function (cdf) of pooled anomaly scores over the first $m$ batches:

$$\hat{F}_m(r) := \frac{\sum_{j=1}^m \sum_{k=1}^{K_j} \mathbf{1}\{R_{\text{anom},k}^{(j)} \leq r\}}{\sum_{j=1}^m K_j},$$

with the convention that a sum over an empty index set is 0. The following theorem deals with the cdf of anomaly scores in this setting; it immediately applies to the cdf of nominal scores too.

**Theorem 1** (Convergence of Pooled Anomalies' Scores)**.** *As $m \to \infty$, the empirical cdf $\hat{F}_m(r)$ converges almost surely to a well-defined cumulative distribution function $F_{\text{pool}}(r)$:*

$$\hat{F}_m(r) \to F_{\text{pool}}(r) := \frac{\mathbb{E}\left[ \sum_{k=1}^{K_1} \mathbf{1}\{R_{\text{anom},k}^{(1)} \leq r\} \right]}{n\tau} \quad a.s.$$

*where the expectation is over both the random batch composition and the stochasticity of the data.*

The proof of Theorem 1 is deferred to the Appendix, along with a corollary extending this result to multiple LODA projections.

**Remark.** For each finite $m$, the function $\hat{F}_m$ is the empirical cdf of the anomaly score of a uniformly randomly chosen anomaly among all anomalies observed in the first $m$ batches. The theorem says that as $m \to \infty$, this empirical cdf converges almost surely to $F_{\text{pool}}$. Thus, $F_{\text{pool}}$ can be interpreted as the limiting cdf of anomaly scores drawn in this way. We conjecture that results in a similar vein can be obtained with batch data for other unsupervised anomaly detectors such as local outlier factor and one class SVM, and likely even for per-batch isolation forest—despite its additional internal randomization process.

## 3 Results

### 3.1 Results roadmap

We briefly describe the `acanag` Python library and practical issues in Section 3.2. In Sections 3.3 and 3.4 we probe the strengths and weaknesses of the AAA method in a series of simulations. In Section 3.5 we test the method on batch data from eight well-known benchmark datasets (five real, one synthetic, two semi-synthetic). We also look at the effect of the choice of, and number of, ensemble anomaly detectors. Finally in Section 3.6 we test the method on real time series data containing real anomalies along with additional synthetic ones—from the GECCO challenge Moritz et al. (2018), showing the method's potential for anomaly detection in time series.

### 3.2 The `acanag` library

The `acanag`[1] Python library contains functions to run the AAA algorithm on data batches from ordered or unordered datasets. Users can choose the number and types of anomaly detectors to include, the supervised classifier, and the active learning strategy. All other parameters have default values that can be modified by the user. A full example notebook is included in the library, as well as reproducible code for all trials found in this paper.

AAA is a fast algorithm. Its run-time basically depends on how quick the classifier and active learner are. As an example, with a random forest classifier and uncertainty sampling providing 5 examples to label per batch, it takes less than 1 second per batch to run AAA on a laptop PC for 100 batches of 10-dimensional data of size 500.

### 3.3 Mixture distributions with varying separability

We tested AAA against three originally LODA-inspired methods: LODA, active-LODA, and GLAD (Pevnỳ, 2016; Das et al., 2016; Islam et al., 2018) which range from aggregated unsupervised anomaly detection with no learning (Pevnỳ, 2016) to active anomaly detection with a proxy neural network (Islam et al., 2018). We implemented these methods in the batch setting in order to compare them directly with AAA, and allowed five data points to be sent to the expert from each batch.

In all simulations, unless otherwise stated we set the number of initial data points to 1000, the proportion of them with already known labels to 0.1 (i.e., 100 pre-labeled of which we enforced 98 to be nominals and two anomalies), the batch size to $B = 500$, the number of batches to 200 (except for active-LODA: 50 or 100 batches due to optimization issues); results were averaged over five runs.

We first considered the setting in which i.i.d. data points $X_i$ were generated from a mixture distribution of nominals and anomalies $f_X \sim (1-\tau)f_{nom} + \tau f_{anom}$ where $f_{nom}$ was the distribution of nominals, $f_{anom}$ the distribution of anomalies, and $\tau = 0.01$ the mixture parameter. Thus, for each data point we performed a Bernoulli trial $\mathcal{B}(0.01)$ and if we obtained 0 we generated one value from $f_{nom}$; otherwise we generated one value from $f_{anom}$. Figure 2 (column 1) shows the component densities of four 2-dimensional data sets simulated in this way with nominals generated from $\mathcal{N}((0,0), I_2)$ and anomalies from $\mathcal{N}((c,c), I_2/10)$, where $c \in \{0.5, 1, 1.5, 2\}$ and $I_2$ is the 2-dimensional identity matrix. We set logistic regression to act as AAA's supervised classifier in these trials, with `class_weight = 'balanced'`, and sent 5 greedy active learning-selected candidates to the expert from each batch. The ensemble models were LODA projections obtained following Pevnỳ (2016) with the constraint of 15 projections at most.

For the three hardest cases ($c \in \{0.5, 1, 1.5\}$), AAA significantly outperformed the other methods both in terms of AUC and cumulative anomalies detected, even though those methods were created specifically for LODA projections. GLAD performed slightly better than AAA in the simplest setting ($c = 2$). Supplementary Figs 2 and 3 in the Appendix provide a visual interpretation of how well each of the four methods separates anomaly scores from nominal scores after aggregation.

We tested several extensions to these results.

- **10-dimensional data**. Plots for the same trials but with 10-dimensional data, with nominals generated from $\mathcal{N}((0,0,\ldots,0), I_{10})$ and anomalies from $\mathcal{N}((c,c,\ldots,c), I_{10}/10)$ for $c \in \{0.5, 1, 1.5\}$, can be found in Supplementary Fig. 4 in the Appendix; conclusions are similar to those above.

- **Other 2-dimensional distributions**. Three further difficult 2-dimensional mixture densities are shown in Supplementary Fig. 5 in the Appendix. In all of them, AAA succeeded in detecting some anomalies, while all three LODA-inspired methods failed.

- **Active learning tradeoff**. Under the original 2-dimensional setting above, we tested the tradeoff of $a \in \{0, 1, 2, 3, 4, 5\}$ active uncertainty sampled points versus $5 - a$ greedily sampled points (also an active learning strategy) sent to the expert in each batch. Increasing $a$ did not systematically

---

[1] https://github.com/yagu0/ActiveAnomalyAggregation

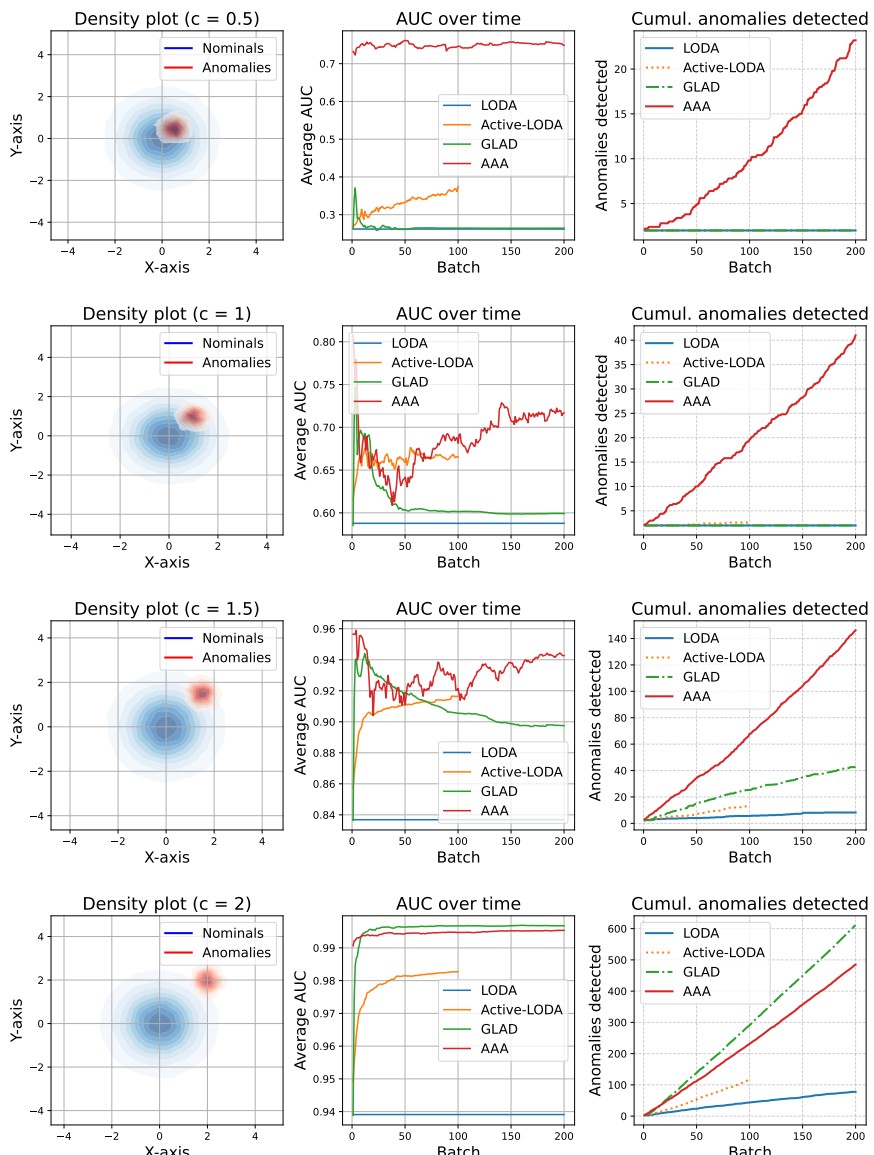

Figure 2: Two-dimensional i.i.d. Gaussian mixtures of nominals and anomalies. AUC and cumulative anomalies detected over time for four methods as new batches arrive. Anomalies occur with probability 0.01. Nominals follow $\mathcal{N}((0,0), I_2)$ and anomalies $\mathcal{N}((c,c), I_2/10)$, with results averaged over 5 trials.

affect the average number of anomalies detected over time (See Supplementary Fig. 6 in the Appendix). The only statistically significant effect was in the easy $c = 2$ discrimination setting where unsurprisingly there was little value in adding an uncertainty sampler to the greedy selector. See the Appendix for more details on the Wilcoxon paired sign rank tests performed.

- **Effect of the choice of supervised classifier**. We ran 5 repeats of the same 2-dimensional trials for each of logistic regression, random forest, and a multilayer perceptron as the classifier. All three classifiers performed well and there was no clear winner; see Supplementary Fig. 7 in the Appendix along with technical details.

- **Effect of the batch size**. It is not immediately clear what influence the batch size $B$ might have on overall performance, over time, of AAA—or the other three methods. It would be of interest to try and understand the influence of $B$ further, but given the number of variables in play over

and above the batch size: data distribution over time, choice of score functions, choice of supervised classifier and its parameters, number of candidates to send to the expert from each batch, choice of active learner, true anomaly frequency), we consider a general answer to this question out of the scope of the current paper.

### 3.4 Score distribution simulations

In this second series of simulations, we take the *score distribution* rather than *data distribution* point of view to show that for AAA, true anomalies' scores need not be high to be informative; they just need to have sufficiently different distributions to nominals.

- **Score distributions**. Four score distributions are presented in column 1 of Fig. 3. It is important to understand that these plots are *not data density plots like in* Fig. 2; they correspond to (simulated) distributions of downstream anomaly and nominal *scores*. In all four trials, two-dimensional i.i.d. Gaussian data were generated. Each data point was then assigned a vector of ten random uniform scores between 0 and 1, corresponding to the scores output from ten random uniform "anomaly detectors". We then artificially labeled each data point as an anomaly or nominal via some *function of the random uniform scores output by the first (of the ten) random score anomaly detectors.* Full details of the simulation setting for each of the four trials are provided in the Appendix. Fig. 3 shows that AAA significantly outperformed the three LODA-based methods on all four score distributions.

- **Time series anomalies**. In time series data, anomalies can be contextual: they may not directly depend on the individual data values at each time point but instead on relationships between data values at different time points. We set the first point $x_1 = (0, \ldots, 0)$ and then randomly generated 10-dimensional Gaussian mixture *jumps* for $i \geq 2$, leading to the following (random walk) data:

$$X_i \sim x_{i-1} + \left(\frac{1-\tau}{2}\right)\mathcal{N}((5,\ldots,5), I_{10}) + \tau\mathcal{N}((5.5,\ldots,5.5), I_{10}/100) + \left(\frac{1-\tau}{2}\right)\mathcal{N}((6,\ldots,6), I_{10}).$$

True anomalies were considered the points generated with "medium-sized jumps" from $\mathcal{N}((5.5,\ldots,5.5), I_{10}/100)$ with probability $\tau = 0.01$. We constructed a diverse ensemble of ten anomaly detectors made up of five LODA projections, isolation forest, one class SVM, local outlier factor, a random score function, and a Euclidean distance score function (which could partially recognize jump magnitudes). All other parameter details can be found in the Appendix. Figure 4 shows what one of the data dimensions looks like (top left), what the score distributions look like for the Euclidean distance score function (bottom left), and the performance of the four methods (top and bottom right). AAA successfully detected anomalies and got better at doing so over time, while the other three methods failed.

### 3.5 Benchmark batch datasets

We tested the iterative batch performance of AAA on the eight benchmark datasets with more than 1000 points from the original isolation forest paper (Liu et al., 2012), of which five were real data, one was simulated, and two were real data with injected synthetic anomalies. We used the same diverse set of ten anomaly detectors from Section 3.4. We treated the eight datasets as unordered data sets and ran consecutive batch trials on five random permutations of each. We then averaged the cumulative anomalies detected over time as new batches arrived. All technical details can be found in the Appendix. Fig. 5 shows that AAA outperformed the other three methods on all eight datasets, sometimes by a large amount.

**Importance of the choice and number of ensemble detectors**. In all of the trials after the initial simulations, we used the same diverse ensemble of ten anomaly detectors. To analyze the effect of the selection and number of detectors, we took the Satellite dataset (see Fig. 5 and the Appendix) and our usual diverse ensemble of ten anomaly detectors and (i) randomly shuffled the data 5 times; then for each shuffle (ii) randomly selected 100 subsets of size $s$ for each $s \in \{1, \ldots, 10\}$ and ran AAA on batches of size $B = 100$,

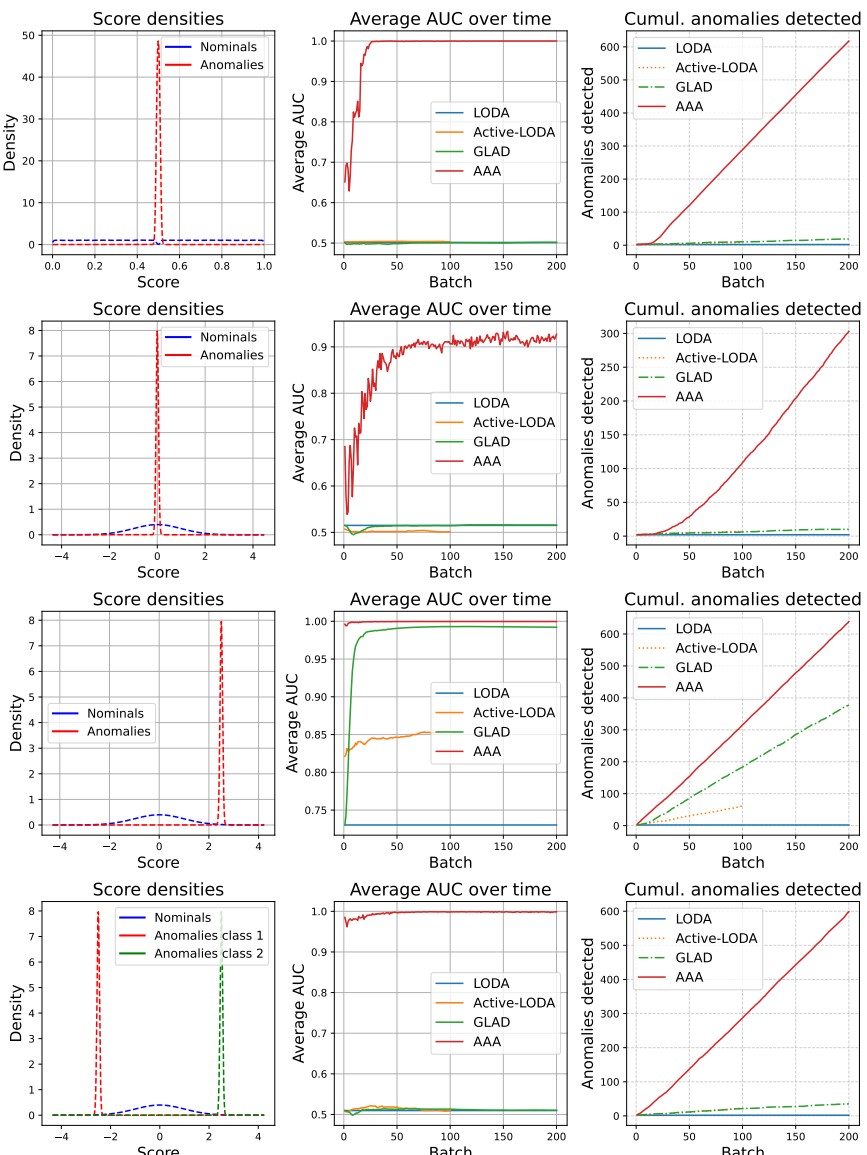

Figure 3: AUC and cumulative anomaly detection performance of LODA, active-LODA, GLAD, and AAA for an ensemble of ten score models. The first score model has the score distribution shown in column 1. The other nine models were uninformative random uniform scores (not shown). The score distributions shown have not been scaled by their relative proportions in the mixture in order to see the (rare) anomaly scores' distributions better. Results were averaged over five trials.

with a random forest classifier and greedy active learning. All subplots in Fig. 6 show the same average cumulative anomalies detected for the ten possible subset sizes as solid lines. For each subset size's subplot, a colored interval shows the 95% range of actual cumulative anomalies detected after a certain number of batches across the 500 values. This shows that certain subsets of $s$ anomaly detectors can perform better over time that certain subsets of $s'$ detectors for $s' < s$. However, for instance, an average subset of size 4 performs better than 95% of subsets of size 1, while 95% of the time, the set of 10 detectors performs better than the average subset of size 4. We also see that the average improvement increases over time but less and less so over time. This suggests that adding even more detectors to the setup would be possible in this example.

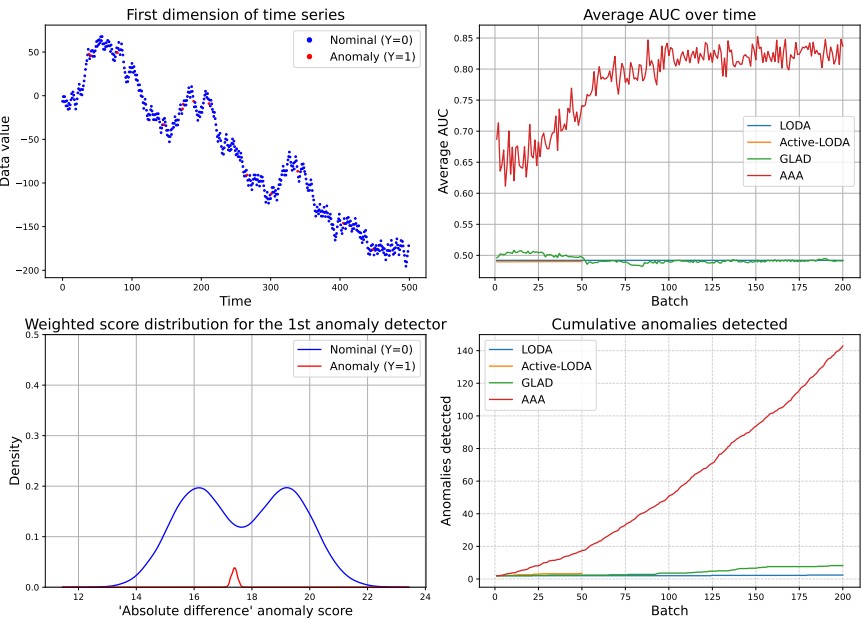

Figure 4: Time series simulations for anomaly detection. Top-left: the data for the first (of ten) dimensions along with anomaly status. Bottom-left: the joint distribution of anomaly and nominal scores for the absolute Euclidean distance score only. Top-right: AUC over time as more batches are added, for four anomaly detection methods. Bottom-right: cumulative anomalies detected.

### 3.6 The GECCO water quality time series

Given we would also like to apply the AAA method to ordered time series data batches, we further tested it on the GECCO challenge dataset (Moritz et al., 2018), which involved 139 566 consecutive data points made up of 9 water-quality related variables associated with anomaly or nominal labels. Real anomaly events were supplemented by artificial anomalies by the challenge organizers. We used the same diverse set of ten anomaly detectors from Section 3.4. Each sequence of contiguous anomalies was replaced by one randomly selected anomaly before running AAA; this random choice was repeated 20 times and results were averaged. Further technical details can be found in the Appendix. Fig. 7 shows the average cumulative number of anomalies detected. On average AAA detected 39 of the 50 remaining anomalies, significantly higher than the number detected by the other three methods. Since the GECCO data was highly non-stationary, it was not a surprise that active-LODA appeared to perform better than GLAD, given that GLAD tried (and therefore failed) to use the raw data to optimize score weights via its neural network. Indeed, GLAD—which is initialized to equal weights—performed similarly to unsupervised LODA.

## 4 Discussion

AAA, the supervised score aggregation method for active anomaly detection presented here, extends active learning-based anomaly detection methods like active-LODA and GLAD to a general supervised framework and to the batch or ordered data setting for arbitrary ensembles of unsupervised anomaly detectors. The main feature differentiating AAA from previous score-based active anomaly detection methods is in how it drops the hypothesis permeating the literature that only individual anomaly score models that output "mostly higher scores for anomalies than for nominals" are informative. Here we show that even "mid-range" scores for true anomalies provide useful discriminatory information as long as there are differences between the distributions of anomaly and nominal scores.

Despite no systematic performance difference between the different classifiers in Supplementary Fig. 7 in the Appendix, we recommend using—in the absence of prior knowledge—random forest when running AAA

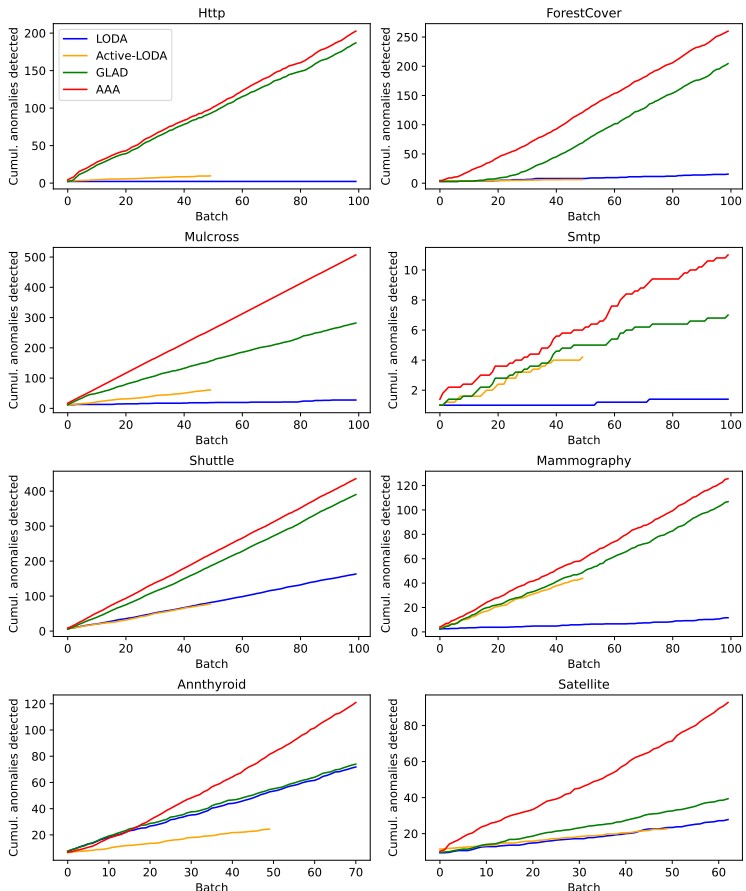

Figure 5: Average cumulative anomalies detected (over 5 trials) in the eight datasets with more than 1000 points in Table III of (Liu et al., 2012).

from the `acanag` library. This is simply because random forest is robust against correlated score functions (unlike logistic regression) and robust—unlike simple neural networks—against ensemble functions which may output wildly different ranges of scores.

As for choosing an active learning strategy when running AAA, it remained inconclusive whether it was worth supplementing greedy active learning (sending the most likely-to-be anomalies to the expert) with another active learner such as uncertainty sampling. When the associated classification task was simple yet unbalanced (Supplementary Fig. 6, top-left), fully greedy active learning was statistically significantly better than full uncertainty sampling. However, in the other three—harder—cases, the trade-off between the two did not systematically suggest concentrating on labeling candidates from one or the other.

As it currently stands, AAA could under-perform if early labeled data suggested that all of the true anomalies' scores were in one zone of the $M$-dimensional score space, whereas in fact there were significant (undetected) concentrations of true anomalies elsewhere in the space. If a decision boundary was learned by AAA's associated supervised classifier on an unrepresentative set of anomalies, this could overtly influence both the set of future predicted anomalies and future active learner-obtained candidates sent to the expert for labeling in subsequent batches. One way to get around this would be to also provide a randomly chosen candidate to the expert from time to time in order to better explore the score space, though if anomalies were extremely rare, it might take a long time for this mechanism to stumble upon anomalies elsewhere in the score space. In any case, if anomalies hidden elsewhere in the score space were not associated with the highest scores from at least one of the score detectors in the ensemble, the three LODA-inspired methods would not find them either.

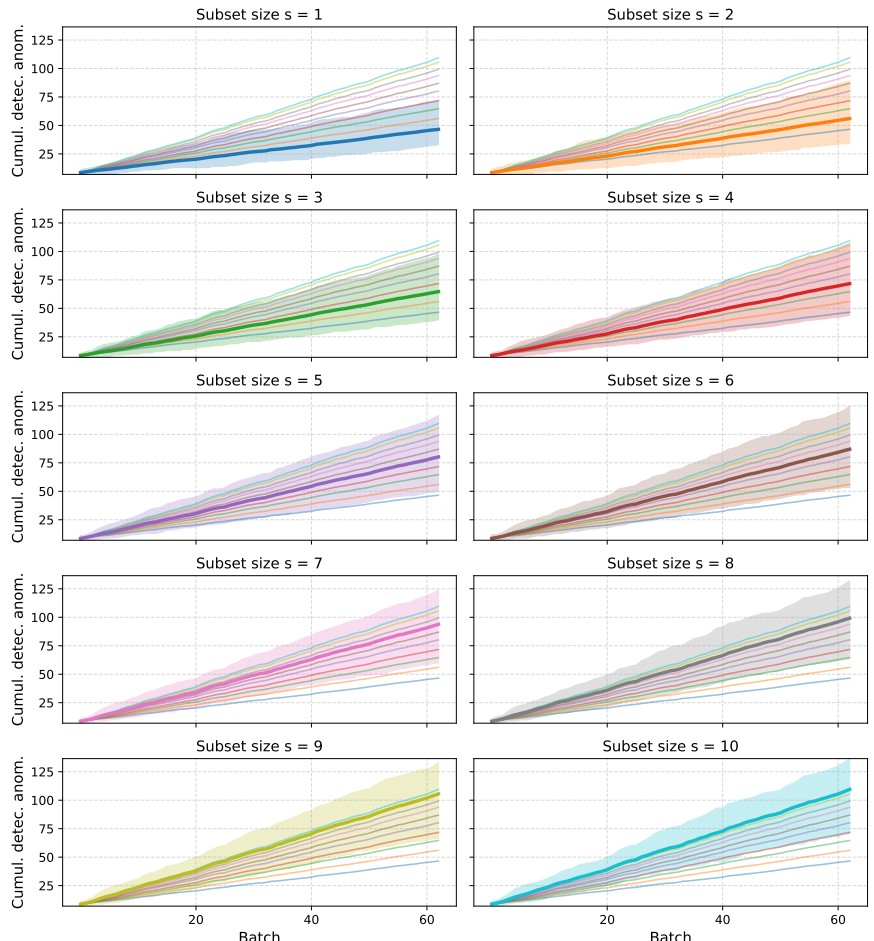

Figure 6: Influence of the size $s \in \{1, \ldots, 10\}$ of the ensemble of anomaly detectors on cumulative anomaly detection performance, for the Satellite dataset (see Appendix for details). Solid lines indicate the average over 500 random subsets of size $s$ from the original ten anomaly detectors. Colored zones indicate the 95% range of values for each choice of $s$ over the 500 trials and across batches.

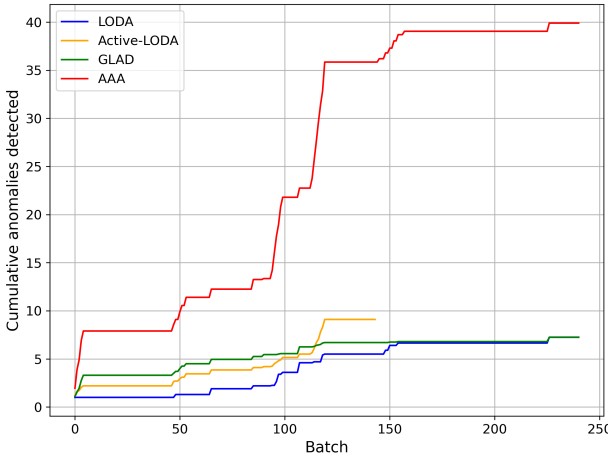

Figure 7: Average cumulative anomalies detected (over 20 trials) in the GECCO time series dataset, with a randomly selected anomaly representative from each of the 51 anomaly intervals included in each trial.

In our simulations, as in previous work (Das et al., 2016; Pevnỳ, 2016) we supposed a relatively high anomaly rate $\tau$ (typically 1 in 100 in our trials), even though this could be much lower in real-world settings. Nevertheless, up to a point, AAA is agnostic with respect to this value, as long as it is still shown—like in the simulations—a small number of useful labeled anomalies and nominals to begin with. Indeed, on the real data from the GECCO industrial challenge (the real anomalies were also supplemented by synthetic anomalies by the organizers), despite an anomaly rate of 0.0004, AAA discovered on average 39 of the 50 anomalies despite only seeing one labeled anomaly to begin with.

## 5 Acknowledgements

This work was supported by the French government under the France 2030 program, as part of the SystemX Technological Research Institute within the Confiance.ai project.

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
