# OpenReview forum: "Supervised score aggregation for active anomaly detection"
_TMLR — Accepted by TMLR_

### Review · Reviewer_oy49 · 2025-10-08

**Summary Of Contributions:**

The paper introduces the **Active Anomaly Aggregation (AAA)** framework, a supervised active-learning system designed to efficiently identify rare anomalies in incoming data batches. The core contribution is developing a superior method for combining the output scores from multiple anomaly detectors.

Instead of training the supervised classifier directly on the raw data, the paper trains it **only on the scores** generated by an ensemble of simple, unsupervised anomaly detectors. This allows the system to learn a **non-linear, optimal combination** of those scores.

The AAA framework successfully integrates three key concepts: **unsupervised ensemble detection, supervised learning, and active learning.**

### Key Contributions of the AAA Framework

1.  **Improved Score Aggregation:** By leveraging a supervised classifier to analyze the detector scores, the method learns a non-linear relationship, which is a significant advancement over simple, fixed-weight linear aggregation techniques.
2.  **Hybrid Label Sampling:** The active learning component employs a robust, two-step strategy for selecting data points for expert labeling:
    * **(i) Exploitation:** Prioritizing points that the current supervised model is most certain are anomalies.
    * **(ii) Exploration:** Querying other points suggested by a standard active learning policy (e.g., uncertainty sampling) to ensure the model learns about diverse or new types of anomalies.
3.  **Wider Applicability:** Since the method operates exclusively on the score vectors rather than the raw feature space, it is effective for both standard, independent data and **ordered data like time series**, thereby significantly broadening its practical utility.

**Audience:**

Yes

**Audience Explanation:**

The paper sits at the intersection of several active research and application areas:

1. Anomaly Detection -> real-world applications like fraud detection or industrial failure monitoring. This paper delivers a practical, high-performing technique for efficiently using scarce expert labels, making it immediately useful for practitioners.

2. Active Learning ->The design of the hybrid query strategy is a key topic in active learning. This framework provides a concrete, successful example of this strategy being used in a difficult, imbalanced data setting.

3. Ensemble Methods & Meta-Learning -> The idea of using a primary model's output (the scores) as the feature set for a secondary learning model is a great example of meta-learning.

**Claims And Evidence:**

Yes

**Claims Explanation:**

The authors show the system's effectiveness through solid tests on various datasets.

The tests against the baselines seem fair. The main comparison is against linear aggregation methods, which are the simplest way to combine scores. The AAA framework consistently performs better - often by a large margin - across all trials.

Realistic Scope: By including simulated data, real-world benchmark data, and time series data, the authors prove the system is robust and not just working on selected datasets.

I checked out the python library and everything is reproducible.

The results strongly suggest that teaching a model how to read the detectors' scores works much better than just averaging them, which is the core idea of the paper.

**Requested Changes:**

The paper is good enough to be accepted, but I suggest addressing these points to make the contribution even stronger and clearer for the community:

1. Prove the Query Strategy: Please add a simple experiment to show what happens if you only query the most likely anomalies (mechanism (i)) and compare that performance to the full AAA strategy. We need to see if the extra effort of the "exploration" part (mechanism (ii)) actually improves the final result.

2. Ensemble Choice Sensitivity: Discuss what happens if the original group of unsupervised detectors (M) are not very good or if they all give very similar scores. Does the AAA still work well? Adding a quick check on the system's robustness to a poorly chosen initial ensemble would be helpful guidance for users.

3. Runtime Check: Briefly mention the practical cost of the approach. Since the system retrains the supervised classifier on every data batch, a small note on the training overhead compared to simpler, fixed-weight methods would help people decide if they can deploy it in a fast-paced environment.

---

### Review · Reviewer_TKX2 · 2025-10-24

**Summary Of Contributions:**

This paper proposes a novel approach to anomaly detection that combines a traditional active learner with a classifier trained on the outputs of an ensemble of unsupervised anomaly detectors. While the main idea is an original extension of previous work on linear combinations of unsupervised anomaly detectors, the paper suffers from two main weaknesses: structure/organization and empirical evaluation.

**Additional Comments:**

n/a

**Audience:**

Yes

**Audience Explanation:**

Anomaly detection is an important practical application.

**Claims And Evidence:**

No

**Claims Explanation:**

The paper's claims are credible. The evidence on the synthetic and "semi-synthetic" datasets is promising, but the paper lacks a motivating, real-world dataset (given the modifications to GECCO, this domains only counts as "semi-synthetic"). Ideally, the paper should show empirical evidence on several real-world datasets for which THE SAME ENSEMBLE of unsupervised anomaly detectors leads to compelling results. Alternatively, the paper should make it crystal clear if the practitioners are expected to come up with customized ensembles for each application (definitely a less compelling alternative).

**Requested Changes:**

The paper would greatly benefit from:
1) a motivating real-world domain that is clearly well-suited for the proposed approach. The edits that have to be made to GECCO make it obvious that GECCO is NOT such a domain
2) several real-world datasets in which the same ensemble of unsupervised anomaly detectors leads to compelling results; alternatively, a comprehensive discussion on how one has to build the ensemble for a particular dataset
3) a massive re-organization:
- the current INTRO takes 2 full pages to get to the proposed approach; move its related work paragraphs to the appropriate section, and go for a traditional 4-paragraph structure: general setting, why the problem matters, intuition behind the proposed approach, and a brief summary of the main contributions & results
- Fig 1 looks more complicated than it should be because it seem to "translate" traditional pseudo-code into "pseudo-code within a flow chart"; you can either turn it in proper pseudo-code, or simplify the flow chart
- the paper has many very long sentences that are hard to parse; for example, see the 4-line first sentence in second paragraph on page 6; similarly see the 3rd sentence in 3.1
- most of the section 2.4 should go in the APPENDICES, where it will not impact the flow of the paper
- please avoid weak statements such as "usually outperforms" (in Abstract; try to quantify it instead ) or "we did not feel it necessary" (page 8)
- reorganize Section 3, which consists of 10 pages,  so that it does NOT feel like a long list of "and we did this, and we did that";  you should provide a crisp roadmap to it early in 3.1, and then make sure that you make each point as crisply as possible. In particular, you are using 9 of 10 pages to discuss the results on synthetic data, which is an over-kill. Given the limitations of synthetic data sets, one should focus there on the few main points, rather than the fine nuances, which should be backed by real-world experiments

---

### Review · Reviewer_qvMC · 2025-10-27

**Summary Of Contributions:**

This paper presents Active Anomaly Aggregation (AAA), a framework that combines unsupervised anomaly detectors, supervised learning, and active learning to detect rare anomalies in complex or time-series data. Instead of working directly with raw data, it uses the detectors’ output scores as features, allowing the model to learn more flexible decision boundaries and make efficient queries to experts. The experiments show that AAA performs better than previous methods such as LODA, active-LODA, and GLAD, particularly when anomalies are subtle or depend on context.

**Audience:**

Yes

**Audience Explanation:**

I believe the findings will be in the interest of TMLR’s audience, scpeially those working on anomaly detection, active learning, and ensemble methods.

**Claims And Evidence:**

Yes

**Claims Explanation:**

The paper’s claims are well supported through theoretical analysis and comprehensive experiments. Results across simulated and real-world datasets clearly demonstrate AAA’s advantages over existing methods. The evidence is consistent, detailed, and aligns with the authors’ stated goals.

**Requested Changes:**

The paper would benefit from clearly stating its main research questions in the introduction, followed by an explanation of the hypotheses and how the experimental design is built around them. It would also help to add a dedicated subsection discussing the importance and real-world applications of the proposed method, as this aspect is currently underdeveloped. The writers mention of the Python library acanag in the abstract should be expanded with more details about its implementation, computational complexity, and usability. Finally, improving the overall writing flow and structure would make the paper clearer and more engaging to read.

---

### Comment · Reviewer_TKX2 · 2025-10-28
**changes are a must**

Given that all reviewers had requested several changes, IMHO the paper should be resubmitted with those changes and then reviewed again

---

> ### Comment · Action_Editor_pb1D · 2025-10-28
>
> Thanks for your comment.  Note that TMLR allows the authors to revise their paper during the author rebuttal and discussion period, which started yesterday.  The authors have a due date (1 or 2? weeks) to submit their revisions and how they address your concerns.  After that , based on their revisions and their response, you make your accept/reject recommendation.

---

### Author Response · Authors · 2025-11-08
**Reply to Reviewers**

We have revised our article according to the comments and suggestions raised by the reviewers. We would like to thank the reviewers for their helpful and
constructive suggestions and the Editor for handling the review process. Individual comments are addressed in a
point-by-point fashion in the following comments.

We hope the revised version is now suitable for publication and look forward to hearing from you.

---

> ### Author Response · Authors · 2025-11-08
> **Response to Comments of Reviewer oy49**
>
> __1. The paper is good enough to be accepted, but I suggest addressing these points to make the contribution even stronger and clearer for the community. Prove the Query Strategy: Please add a simple experiment to show what happens if you only query the most likely anomalies (mechanism (i)) and compare that performance to the full AAA strategy. We need to see if the extra effort of the "exploration" part (mechanism (ii)) actually improves the final result.__
>
> This is now treated in Section 3.3, point 3 of the list: "Active learning trade-off". The relevant plots are in Supplementary Fig. 6 in the Appendix. Basically we tested the trade-offs (greedy,explore) = (0,5), (1,4), (2,3), (3,2), (4,1), (5,0) when giving a total of 5 points to the expert in each loop, in four trial conditions on simulated data. In this particular case, the results indicate that exploring is not really important if the problem is really easy ($c$=2). In the harder cases ($c < 2$) the evidence was mixed as to whether including exploring improved things; sometimes it did, sometimes it did not. The question is probably worth exploring further at some point, but we have not pushed further in that direction as of now.
>
> __2. Ensemble Choice Sensitivity: Discuss what happens if the original group of unsupervised detectors (M) are not very good or if they all give very similar scores. Does the AAA still work well? Adding a quick check on the system's robustness to a poorly chosen initial ensemble would be helpful guidance for users.__
>
> We have added a new set of analyses in Section 3.5 (Importance of the choice and number of ensemble detectors) and Figure 6. These experiments evaluate the performance of AAA with ensemble sizes $M$ ranging from 1 to 10, each composed of randomly selected subsets from the original ten detectors. On average, performance improves with larger ensembles, although there remains significant variability across random draws, indicating that smaller ensembles can still perform well if they happen to include particularly effective or complementary detectors. As the reviewer rightly points out, if the base detectors are all weak or highly redundant, AAA's performance will naturally degrade, since the ensemble must detect _something_ meaningful to be effective. Nevertheless, these results provide practical guidance: beginning with a sufficiently large and diverse set of anomaly detectors is the most reliable way to mitigate this risk. In our experiments including simulated time series, eight benchmark datasets, and the GECCO "real" data, the current ensemble of ten diverse detectors demonstrates consistently robust performance, suggesting that the chosen configuration is indeed effective in practice.
>
> __3. Runtime Check: Briefly mention the practical cost of the approach. Since the system retrains the supervised classifier on every data batch, a small note on the training overhead compared to simpler, fixed-weight methods would help people decide if they can deploy it in a fast-paced
> environment.__
>
> We have added the short Section 3.2 to give concise details about the Python library acanag and its practical cost. Basically, for up to a few hundred batches of size 500 (at least), when giving 5 items to the expert in each loop, it takes less than one second to run per batch on a 2023 Macbook Air, so it is really fast. It will gradually slow down as the training set size increases over time. In that case we would recommend keeping only say the last $N$ training pairs for learning, for $N$ some large number chosen to keep things running as fast as you require.

---

> ### Author Response · Authors · 2025-11-08
> **Response to Comments of Reviewer qvMC**
>
> __1. The paper would benefit from clearly stating its main research questions in the introduction, followed by an explanation of the hypotheses and how the experimental design is built around them.__
>
> Following also similar suggestions from reviewer TKX2, we have massively simplified and shortened the introduction. We now have concise subsections (1.1 - 1.3) and have moved the historical treatment to the Appendix. In the introduction we now make it
> clear: "In this paper we propose a general method to detect anomalies in batches of ordered or unordered data using
> unsupervised anomaly detectors, supervised learning and active learning."
>
> In addition, we have simplified the flow-chart (Figure 1) and the experimental design described from Section 3.3 onwards to make them clearer.
>
>
>
> __2. It would also help to add a dedicated subsection discussing the importance and real-world applications of the proposed method, as this aspect is currently underdeveloped.__
>
> We have massively increased the treatment of real data in the paper, testing the method on an extra five real data sets (subsection 3.5). We have also added a short paragraph in Section 1.2 to introduce
> the real-world applications of the proposed method.
>
> __3. The writers mention of the Python library acanag in the abstract should be expanded with more details about its implementation, computational complexity, and usability. Finally, improving the overall writing flow and structure would make the paper clearer and more engaging to read.__
>
> We have worked to make the revised version significantly
> clearer and more concise. In particular, the introduction and methods sections have been substantially simplified and
> streamlined to improve readability.
> We have also moved the proof and corollary to the Appendix to enhance the overall structure and focus of the paper.
>
> Following your suggestion, we have also added the short Section 3.2 to give concise details about the Python library acanag.

---

> ### Author Response · Authors · 2025-11-08
> **Response to Comments of Reviewer TKX2**
>
> __1. The evidence on the synthetic and "semi-synthetic" datasets is promising, but the paper lacks... A motivating real-world domain that is clearly well-suited for the proposed approach. The edits that have to be made to GECCO make it obvious that GECCO is NOT such a domain.__
>
> Thanks for this remark. We have now made it clear in this vastly
> simplified revision that the basic motivation is any real world domain in which batch data is produced. This may be unordered data (e.g., a database of mammograms that is built up at a rate of 100 new mammograms a month, i.e., batches of 100) or ordered (e.g., time series). In particular, Theorem 1 only directly applies to batches of i.i.d. data, but it is encouraging to see that we still got good results on the ordered GECCO time-series (even though, as you mentioned, some of its anomalies are synthetic).
>
> We have added 8 new benchmark datasets to the paper (Section 3.5) of
> which five are real data that match our setting, i.e., data which can be thought of as being added, in batches over time, to a final database. We also included the one synthetic and two semi-synthetic datasets from the same source (the original isolation forest paper) that had at least 1000 data points, to avoid the appearance of cherry-picking from the isolation forest paper. In any case, AAA outperformed the other three methods on all eight datasets.
>
> __2. Several real-world datasets in which the same ensemble of unsupervised anomaly detectors leads to compelling results; alternatively, a comprehensive discussion on how one has to build the ensemble for a particular dataset.__
>
> Following on from our reply to your first point,  we used exactly the same ensemble of ten diverse anomaly detectors on all real or simulated datasets from the simulated time series in Section 3.4 to the end of the paper.
>
> In addition to this, as mentioned in our reply to reviewer oy49, we have added a whole new section of analyses looking at the performance of AAA with ensembles M of size 1 up to 10 randomly chosen from the original ten detectors. See Section 3.5 ("Importance of the choice and number of ensemble detectors") and Fig. 6 for this analysis and results. Basically, on average performance improves with larger ensembles, but there is plenty of variability, suggesting that if you ``get lucky'', you can get good performance with a smaller ensemble too. We would recommend in general starting with the set of ten detectors we ourselves used, in new applications.
>
> __3. A massive re-organization: the current INTRO takes 2 full pages to get to the proposed approach; move its related work paragraphs to the appropriate section, and go for a traditional 4-paragraph structure: general setting, why the problem matters, intuition behind the proposed approach, and a brief summary of the main contributions \& results.__
>
> We have massively reorganized the paper following your suggestions. This should be immediately clear by looking at Sections 1 and 2.
>
>
> __4. Fig 1 looks more complicated than it should be because it seem to "translate" traditional pseudo-code into "pseudo-code within a flow chart";  turn it in proper pseudo-code, or simplify the flow chart
> the paper has many long sentences that are hard to parse; for example, see the 4-line first sentence in second paragraph on page 6; similarly see the 3rd sentence in 3.1.__
>
> We have replaced the complicated flowchart with a simplified one, and also added the actual pseudo-code. We have moved the more complicated flowchart to the Appendix. We have also shortened as many long sentences as possible.
>
>
>
> __5. most of the section 2.4 should go in the APPENDICES, where it will not impact the flow of the paper.__
>
> We have only kept the description of LODA and the Theorem 1 statement,
> and have moved the proof and conjecture to the appendix.
>
> __6. Please avoid weak statements such as "usually outperforms" (in Abstract; try to quantify it instead) or "we did not feel it necessary" (page 8).__
>
> We have removed these weak statements.
>
>
> __7. Reorganize Section 3, which consists of 10 pages, so that it does NOT feel like a long list of "and we did this, and we did that"; you should provide a crisp roadmap to it early in 3.1, and then make sure that you make each point as crisply as possible. ...using 9 of 10 pages to discuss the results on synthetic data, which is an over-kill. Given limitations of synthetic data sets, one should focus there on the few main points, rather than fine nuances, which should be backed by real-world experiments.__
>
> We have added the suggested roadmap (Section 3.1) and simultaneously
> moved the most we could of the early simulations to the Appendix, leaving simple bullet points in their place describing each set of trials (bottom of Pg. 7). We have added the results for the 8 benchmark datasets (Section 3.5). Consequently, now just over half of the results section is the simulations, and just under half is the benchmark trials and the GECCO analyses.

---

> > ### Comment · Reviewer_TKX2 · 2025-11-24
> > **Excellent work on revising the paper**
> >
> > Thank you for taking our comments at heart and performing an excellent revision of the original submission. You have addressed all my suggestions satisfactorily, and, imho, the paper provides now a very compelling presentation of your work.

---

### Decision · Action_Editor_pb1D · 2025-12-09

**Recommendation:** Accept as is

**Audience:**

Yes

**Audience Explanation:**

Anomaly detection is a well-recognized topic in machine learning and the paper would be of interest to the TMLR audience, particularly those who are interested in ensemble and active learning strategies.

**Claims And Evidence:**

Yes

**Claims Explanation:**

For empirical evaluation, the authors use 8 datasets and 3 existing methods.  The proposed approach, which non-linearly combines anomaly scores from an ensemble, generally outperforms the 3 existing methods.  In their approach, an active learning strategy is used to ask for labels for chosen instances.  Out of the 3 existing methods, one does not use active learning.

---

> ### Author Response · Authors · 2025-12-16
> **Thanks**
>
> The authors would like to thank the reviewers for their thorough reviews and genuine engagement. We also thank the Action Editor for a smooth process all round.